## Overview Review

precision oncology; precision medicine; structural racism; critical race theory; health equity; racial discrimination; racial bias; cancer disparities

**Corresponding author:**
Lester D. Geneviève,
E-mail: lester.genevieve@unibas.ch

# Impact of structural racism on inclusion and diversity in precision oncology: A scoping and critical review of the literature

Lester D. Geneviève[1] , Bernice S. Elger[1,2] and Tenzin Wangmo[1]

[1]Institute for Biomedical Ethics, University of Basel, Basel, Switzerland and [2]University Center of Legal Medicine, University of Geneva, Geneva, Switzerland

## Abstract

Inclusion and diversity in precision oncology are essential in reducing cancer disparities among racial and ethnic groups. However, present studies have favored the recruitment and participation of Whites, with limited applicability of their results to minority groups. Many reasons for their underrepresentation are downstream manifestations of structural racism. Therefore, this scoping review provides a precise mapping of recruitment and participation barriers for minorities in precision oncology that are associated with structural racism, including a critical appraisal of how disciplinary norms, paradigms, and tools used therein could inadvertently contribute to unforeseen inclusion and diversity challenges. Empirical and theoretical publications from Web of Science and PubMed were searched and analyzed to identify recruitment and participation barriers for minorities in precision oncology. In addition, using the public health critical race praxis (PHCRP) as guiding analytical framework, empirical studies were analyzed to identify unforeseen barriers resulting from simplification processes, assumptions, norms, paradigms, and tools used during the research process. One-hundred thirty-five barriers to recruitment and participation were identified or reported in included publications. They were subsequently categorized as being a manifestation of one of the following forms of racism, namely internalized, interpersonal, institutional, and structural racism. The PCHRP analysis revealed four additional factors to be considered in precision oncology studies in ensuring appropriate representation of their study populations. Future interventions aimed at reducing health disparities should focus predominantly on barriers associated with structural and institutional racism, which should then have ripple effects on other forms of racism. Importantly, the four factors identified through the PHCRP framework could further explain the lower participation rates of minorities in precision oncology and related activities. Therefore, they should be given due consideration by all stakeholders involved in the precision oncology ecosystem, from researchers and healthcare professionals to policy-makers, research ethics committees, and funders.

## Impact statement

This scoping review provides a precise mapping of recruitment and participation barriers that racial and ethnic minorities are likely to face in precision oncology and related activities. It provides useful actionable insights for researchers, clinicians, healthcare and research institutions, and policy-makers to identify how systemic discriminatory pathways can lead to cancer disparities along racial and ethnic lines in precision oncology. In addition, it is the first scoping review to utilize the public health critical race praxis (PHCRP) – an analytical framework rooted in critical race theory – to uncover how assumptions, simplifications, norms, and paradigms used by researchers in their respective studies can further explain the limited representation of minorities. For instance, the PHCRP analysis revealed that researchers tend to embrace a monolithic view of racialized populations, wrongly equating self-reported race to the presence of specific genetic variants or homogeneous genetic backgrounds within these populations. Such a view was also found to be primed by inherent limitations of certain tools used commonly in genomic studies and precision oncology, for instance, publicly available genomic databases missing genetic ancestral information and failing to capture the genetic diversity of populations represented therein. Other factors revealed by the PHCRP analytical framework include the *Black–White binary paradigm*, researchers *ignoring the contribution of racism to observed cancer disparities,* and the *lack of conceptual clarity on the meaning and use of race and ethnicity.* Although predominantly centered on U.S. studies and publications, results of this scoping review and of the PHCRP analysis are likely to be useful to the broader scientific community at tackling the lack of inclusion and diversity in their respective projects since structural racism knows no borders.

**Social Media Summary**
Summary of article: A scoping and critical review of recruitment and participation barriers linked to structural racism in precision oncology.

## Introduction

Inclusion and diversity in precision oncology have been heralded as essential means in addressing cancer disparities between racial and ethnic groups (Rajagopal and Olopade, 2020; Aldrighetti et al., 2021). To this aim, genome-wide association studies (GWASs) have been crucial in identifying genetic variants and genomic regions associated with an increased carcinogenic risk (Sud et al., 2017). However, the majority of GWASs have been conducted on samples from European-ancestry populations, with limited transferability of their results to others (Martin et al., 2017). Despite the alarm raised in the last decade on the unacceptable lack of diversity in GWASs (Popejoy and Fullerton, 2016; Sirugo et al., 2019), attempts at challenging the *status quo* were largely unsuccessful (Fatumo et al., 2022). In 2021, according to the *GWAS Diversity Monitor* (Mills and Rahal, 2020), participants of European ancestry still comprised 97.43% of GWASs for cancer, followed by those of Asian (2.51%), African (0.04%), Hispanic/Latin American (0.01%), and other/mixed ancestries (0.02%; GWAS Diversity Monitor, 2022). It was foreseeable that this lack of diversity would also be precipitating into precision oncology studies. Indeed, Aldrighetti and colleagues (Aldrighetti et al., 2021) investigated the ethnic and racial representation of participants in 93 precision oncology clinical studies. They found that there was an overrepresentation of non-Hispanic Whites (82.3%) in all studies, whereas Hispanic and American Indian/Alaska Native participants were underrepresented (3.4% and 0.3%, respectively; Aldrighetti et al., 2021).

The reasons for low enrolment of racial and ethnic minorities in health research or in biospecimen donation is multifactorial. They include socioeconomic constraints, fear of discrimination from health insurers, mistrust, cultural and language barriers, stigmatization, restrictive research design of specific studies, the lack of awareness, or simply minorities not being invited to participate (Wendler et al., 2005; George et al., 2014; Sharrocks et al., 2014; Lee et al., 2019). These factors have subsequently contributed to the exclusion of minorities in cancer research (Winkfield, 2020; Roberson, 2022). Some of these factors are the result of structural or systemic discrimination, which has been cemented in our contemporary society by racialization processes and inferior societal positions imposed to racial minorities for centuries (Valdez and Golash-Boza, 2017; Adigbli, 2020; Geneviève et al., 2020). These racial categories persist to this day, and it is therefore important to better comprehend the differences between race, ethnicity, and genetic ancestry, and their subsequent utility for precision oncology. Indeed, although being distinct concepts, they are often collected and used variably in precision oncology and other related activities (Bonham et al., 2018; Adigbli, 2020; Popejoy et al., 2020; Krainc and Fuentes, 2022).

Race and ethnicity are *socially* constructed and *fluid* attributes (Senior and Bhopal, 1994; Adigbli, 2020). Their *fluidity* means that these concepts keep changing over time depending on the context and epoch, or how the person self-identified or is racially identified by others (e.g., Italian immigrants in the United States were first considered to be non-Whites, before becoming 'Whites' when they differentiated themselves from African Americans; Saperstein and Penner, 2012; Davenport, 2020). The fundamental distinction between racial and ethnic groups can be resumed as follows: racial groups are 'imposed externally to justify the collective exploitation of a people and are maintained to preserve status differences', whereas ethnic groups are considered as having 'a primarily sociocultural foundation, and [...] exhibited tremendous malleability in terms of who belongs' (Bonilla-Silva, 1997). Indeed, race focuses on observable physical characteristics that have been used to differentiate and discriminate between Whites and non-Whites, between the haves and have-nots, and between alleged superior and inferior races (Braveman and Parker Dominguez, 2021).

In contrast, genetic ancestry is a *biological* attribute that 'involves the comparison of a large number of DNA variants measured in an individual with the frequencies of these variants in reference populations sampled from across the world' (Jorde and Bamshad, 2020; Krainc and Fuentes, 2022). With regard to precision oncology, genetic ancestry data were found to be particularly important for the design of genetic studies (Rajagopal and Olopade, 2020). Carrot-Zhang and colleagues investigated associations between genetic ancestry and different variables (e.g., microRNAs expression, DNA methylation, and somatic alterations) in 10,678 patients and 33 cancer types from *The Cancer Genome Atlas* (TCGA). Their study revealed not only the important confounding effect emanating from technical artifacts associated with genetic ancestry and different cancer subtypes – highlighting the need for these parameters to be given due consideration in genetic studies – but also the need to cater for the limited representation of non-European ancestries (Carrot-Zhang et al., 2020; Rajagopal and Olopade, 2020).

Therefore, with the advent of precision oncology, this further calls for an in-depth assessment of the role that structural racism plays therein (Roberson, 2022), in particular if precision oncology activities are likely to worsen health disparities by not offering equal opportunities of recruitment and participation to all cancer patients (Geneviève et al., 2020). This is an important societal endeavor to not only achieve health equity, but also fight the re-emergence of scientific racism that would attribute health disparities to the unfounded innate biological or genetic inferiority of minority races, instead of critically reflecting and acting on how systemic oppression and discrimination have led to worse health outcomes for minorities (Matthew, 2019). In that regard, structural racism can be understood as to how racialization processes are deeply embedded in the functioning of our societies and dictate how privileges and opportunities are sidetracked to the majority group at the detriment of minorities (The Aspen Institute Roundtable on Community Change, 2021). Recognizing the strong impact structural racism has had up-to-now on diversity in biomedical research and its threat to health equity, the U.S. National Institutes of Health (NIH) responded by launching the UNITE initiative in 2021. The UNITE effort not only focuses on reducing health disparities, but also tackles the long-standing structural problem of limited diversity in the biomedical workforce (Collins et al., 2021). Indeed, low diversity in the biomedical workforce is also another known contributor to lower participation rates of minorities in research (George et al., 2014).

Given the currently observed low participation rates of racial and ethnic minorities in precision oncology and related activities (e.g., genomic studies) and the urgent need to promote health equity, it is paramount to have an in-depth understanding on how structural racism is potentially impacting the recruitment and participation of racial and ethnic minority groups in precision oncology. Therefore, the objective of this scoping review is to offer a

precise mapping of recruitment and participation barriers of minorities in precision oncology activities that are likely to be associated with structural racism. We further aim to carry out a critical appraisal of how disciplinary norms, paradigms, and tools used in precision oncology and related activities could inadvertently contribute to additional and unforeseen barriers for minorities in ensuring appropriate representation.

## Methodology

A twofold methodological approach is used for this study. The first phase consists of a scoping review that follows the methodological framework by Arksey and O'Malley (2005) while abiding to the PRISMA extension for scoping reviews (Tricco et al., 2018). The aim is to answer the following research question:

> What is currently known from the literature on recruitment and participation barriers in precision oncology that are potentially associated with structural racism?

In that regard, we define *recruitment* as the contact made by researchers or healthcare professionals with potentially interested individuals or patients as a prelude to their *participation* in precision oncology activities, whereas *participation* refers to the opportunity given and the ability of research participants or patients to fully and equitably take part in and benefit from precision oncology activities without being unfairly discriminated because of their racial, ethnic, or cultural identities.

The second phase consists of a critical appraisal of how knowledge is generated in the included empirical studies using the *public health critical race praxis* (PHCRP) framework. Here, we identify how disciplinary characteristics, paradigms, and available tools could implicitly influence and bias the recruitment and participation of minority groups in these studies (Ford and Airhihenbuwa, 2010; Ford, 2016). The methodology is explained further in the critical analysis section. The PHCRP was chosen for this study for the following reasons: (i) it is an adaptation and application of critical race theory (CRT; Bridges et al., 2017) to the field of public health, and it centers on health equity research (Ford and Airhihenbuwa, 2010). Therefore, it is highly relevant to investigate the racialization processes induced by structural racism that would contribute to recruitment and participation barriers in precision medicine (Bayer and Galea, 2015; APHA, 2020); and (ii) it allows a robust exploration of the root causes of health inequities that many conventional frameworks do not allow, that is, 'mov[ing] beyond merely documenting health inequities toward understanding and challenging the power hierarchies that undergird them' (Ford and Airhihenbuwa, 2010).

### Search strategy and study selection

Two electronic bibliographic databases, namely PubMed and Web of Science (core collection), were systematically searched on September 9, 2021 and repeated on May 3, 2022 for publications having components and principles of precision oncology/precision medicine and racism/implicit bias. A search strategy was developed for each database with search terms linked by Boolean operators, and they consisted of two main concepts: (i) *structural racism/ racism and implicit bias*, and (ii) *precision oncology/precision medicine.* We included specifically 'implicit bias' as an integral component to structural racism, since its manifestation and internalization in society are due to how structural racism has shaped, over the years, societal culture, policies, practices, and the distribution of socioeconomic opportunities along racial and ethnic lines (Osta and Vasquez, n.d.). The search strategies did not include the concepts of *participation* and *recruitment* since their inclusion as obligatory components drastically reduced the number of publications retrieved per database. Moreover, we wanted our search strategy to be broad enough to capture potentially upstream and neglected barriers that could have an influence on these two concepts. Our latter choice for a wide and comprehensive search strategy is also recommended by the methodological framework used (Arksey and O'Malley, 2005). The search strategy used for each database is listed in Table 1.

The inclusion criteria for the scoping review were: (i) publications have to involve or discuss different aspects and principles of precision oncology (e.g., molecular characterization of tumors and genomic testing) or general aspects of precision medicine that would apply to precision oncology, (ii) include discussions on racial or ethnic minorities, (iii) the publication language of studies is either English or French, and lastly (iv) the publication year ranges from 2010 onward. There were no restrictions on the cancer type being investigated, racial/ethnic minority populations under study, particular branch of precision oncology involved, methodological characteristics of studies (e.g., qualitative, quantitative, or mixed methods), or the nature of publications (empirical or theoretical). Our choice to include nonempirical publications is also guided by the PHCRP framework, where it is recommended to 'draw on empirical data as well as other kinds of information as needed to address each focus's purpose' (Ford and Airhihenbuwa, 2010). Indeed, the barriers

**Table 1.** Search strategies used for PubMed and Web of Science with Boolean operators

| Key concepts | Concept#1: Structural racism and implicit bias | Concept#2: Precision oncology |
|---|---|---|
| PubMed (all fields) | 'structural racism' OR 'systemic racism' OR 'institutional racism' OR 'racial discrimination' OR 'racial prejudice' OR 'racial bias' OR 'ethnic bias' OR 'racial–ethnic bias' OR 'implicit bias' OR 'unconscious bias' OR 'implicit social cognition' OR 'covert racism' OR 'Racism' [Mesh] | 'Precision Oncology' OR 'Personalized Oncology' OR 'Personalized Medicine' OR 'Molecular Oncology' OR 'Molecular targeted therapy' OR 'Precision Medicine' [Mesh] OR (('Next-generation sequencing' OR genomic OR exome OR transcriptome OR nucleotide) AND (cancer* OR oncology OR tumor OR tumour OR malignan* OR neoplasm)) |
| Search Strategy PubMed (all fields): Concept#1 AND Concept#2 | | |
| Web of Science (core collection) | 'structural racism' OR 'systemic racism' OR 'institutional racism' OR 'racial discrimination' OR 'racial prejudice' OR 'racial bias' OR 'ethnic bias' OR 'racial–ethnic bias' OR 'implicit bias' OR 'unconscious bias' OR 'implicit social cognition' OR 'covert racism' OR Racism | 'Precision Oncology' OR 'Personalized Oncology' OR 'Personalized Medicine' OR 'Molecular Oncology' OR 'Molecular targeted therapy' OR 'Precision Medicine' OR (('Next-generation sequencing' OR genomic OR exome OR transcriptome OR nucleotide) AND (cancer* OR oncology OR tumor OR tumour OR malignan* OR neoplasm)) |
| Search Strategy Web of Science (core collection): Concept#1 AND Concept#2 | | |

identified in nonempirical publications cross-validated those identified in the empirical ones while providing some contextual information for a better understanding of the racialization processes leading to recruitment and participation barriers. However, for quality assurance of gathered data, only peer-reviewed publications (both theoretical and empirical) were included in this review.

### Data charting

The retrieved publications were imported into a reference manager software, namely EndNote™ X9 (EndNote, 2021). Automatic and manual duplicate searches were performed. After duplicates were removed, title-and-abstract and full-text screening were performed. Data charting was carried out using a standard data extraction form developed for this study. Data elements charted included: (i) name(s) of author(s), (ii) title of publication, (iii) publication language, (iv) country/countries where the study was carried out, (v) publication type (theoretical or empirical), (vi) study objective(s), (vii) racial/ethnic minority groups involved, (viii) precision oncology branch, (ix) cancer type being investigated, and (x) identified barriers. The identified barriers were then categorized as being a manifestation of one of these four levels namely: *internalized racism*, *interpersonal racism*, *institutional racism*, and lastly, *structural racism* (Jones, 2000; The Aspen Institute Roundtable on Community Change, 2021). However, it is important to note that structural racism is the most profound form of racism, being the starting point of all other forms of racism (Lawrence and Keleher, 2004).

### Critical analysis using PHCRP

The PHCRP (Ford and Airhihenbuwa, 2010) was used and adapted as an analytical framework to critically assess the knowledge generated in the included empirical studies, by paying particular attention to assumptions or simplifications made during the course of the research process that could influence the recruitment and participation of racial and ethnic minorities. The process is centered around four methodological foci, namely, (i) *contemporary race relations*, (ii) *knowledge production*, (iii) *conceptualization* and *measurement*, and lastly (iv) *action* (see Ford and Airhihenbuwa, 2010, for an in-depth explanation). An analysis was then conducted to identify data resulting from simplification processes, assumptions made by researchers, and disciplinary norms/tools that could influence the recruitment and participation of racial and ethnic minorities in precision oncology activities.

### Self-reflexivity and race consciousness

The critical analysis process using PHCRP, that is, the four foci, has been carried out through the lens of *race consciousness*, a principle considered to be the 'backbone of PHCR[P] because it is difficult to investigate racism's contribution to health inequities without first acknowledging and understanding racialization' (Ford and Airhihenbuwa, 2010). With this approach, we recognize the centrality of racialization processes in the differential treatment and outcomes of racial minorities in the gathered literature, and how our racial identity and prior assumptions can also influence the interpretation of our study findings. In this regard, L.D.G. and T.W. are members of different racial and ethnic minorities in Switzerland, whereas B.S.E. is a member of the majority group. Therefore, we are an ethnically

diverse and interdisciplinary research team with expertise ranging from public health, medicine, gerontology, research methodology to ethics, and health law, and we are familiar with some of the health equity issues raised by structural racism in precision health approaches (Geneviève et al., 2020; Geneviève et al., 2022). During the conduct of the PHCRP analysis, we were only able to draw on some of the PHCRP principles (Ford and Airhihenbuwa, 2010) to analyze textual information since we were not involved in the design and planning stages of the included empirical studies. However, to cater for some limitations, we also borrowed CRT-related concepts affecting the generation (and the implications) of knowledge produced by the empirical studies (e.g., the Black–White Binary; Delgado and Stefancic, 2017a). To reduce the influence of any prior assumption, we questioned the validity and applicability of the PHCRP results to precision oncology, and ultimately reached consensus on the four factors identified.

## Results

### Search results

Fifty-five publications were initially retrieved from the search strategies used, of which four were duplicates. Fifty-one publications underwent the title-and-abstract screening phase, during which an additional 25 articles were excluded. Among the eligible 26 publications for full-text screening, two could not be retrieved, making a total of 24 publications were included for full-text screening. Of these, 11 did not meet the inclusion criteria and were excluded. Reference screening of the 13 included publications led to the identification and inclusion of an additional 19 publications, raising the total number of included studies for this scoping review to 32. The identification, screening, and inclusion of studies (and reasons for exclusion) are shown in the following PRISMA flow diagram (Page et al., 2021; Figure 1).

### Characteristics of included studies and racial/ethnic minority populations

Among the 32 publications included in this scoping review, 17 were of an empirical nature and 15 were theoretical (Table 2). Among the empirical studies, 14 were carried out solely in the United States (82.3%). The minority racial/ethnic groups represented in the 17 empirical studies are as follows: Blacks/African Americans ($n = 17$, 100.0%), Hispanics/Latinos ($n = 10$, 58.8%), Asians ($n = 6$, 35.3%), American Indians/Alaska Native ($n = 4$, 23.5%), Native Hawaiians/other pacific islanders ($n = 2$, 11.8%), and mixed ($n = 2$, 11.8%). In U.S.-based studies or international studies using U.S. datasets, it is unclear why researchers often made a distinction between 'Blacks' and African Americans. The characteristics of included publications are summarized in Table 2.

### Recruitment and participation barriers

A total of 135 barriers to participation and recruitment of racial and ethnic minorities in precision oncology and related activities were identified or reported. Barriers emanated principally from structural racism ($n = 53$, 39.3%), followed by institutional racism ($n = 44$, 32.6%), internalized racism ($n = 27$, 20.0%), and lastly, interpersonal racism ($n = 11$, 8.1%). The barriers associated per type of racism are shown in Table 3. More information on the number of

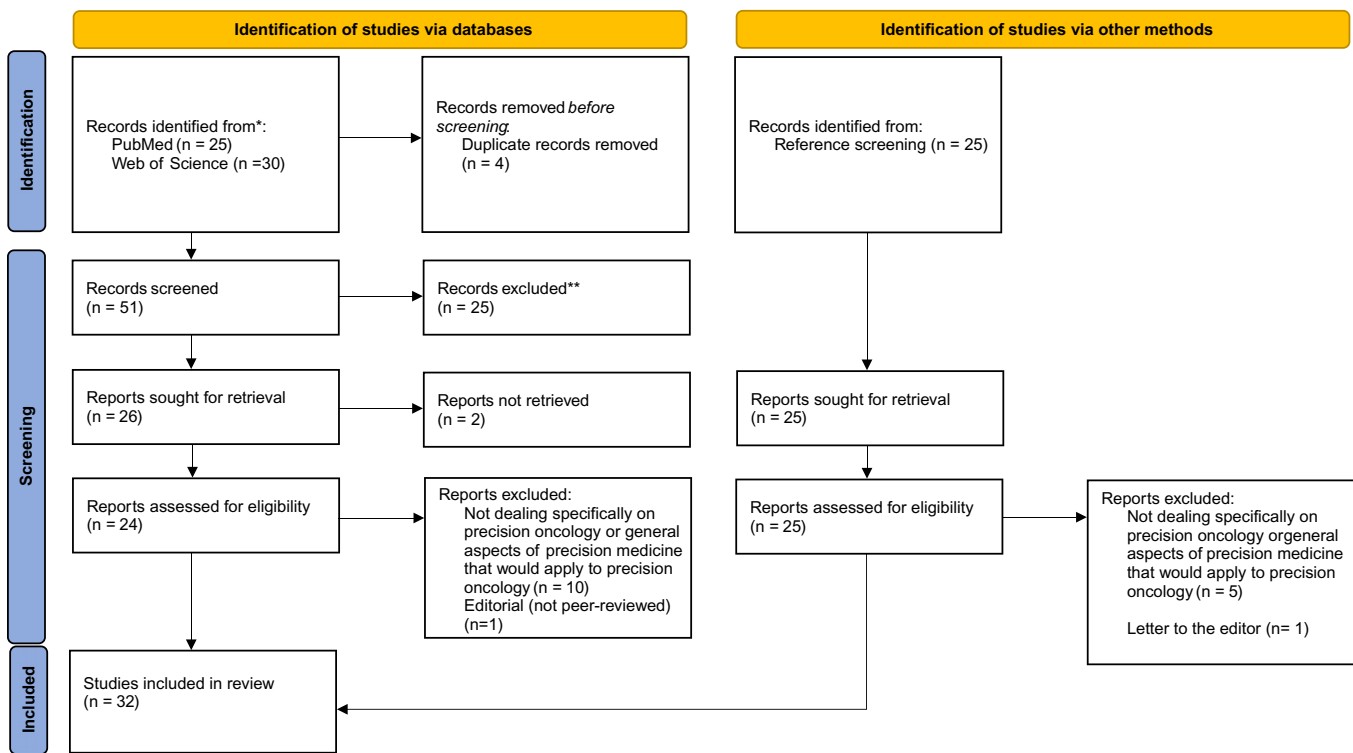

**Figure 1.** PRISMA flow diagram showing the screening and selection processes of included publications.

barriers identified or reported per cluster and per study type is provided in Table 4.

### Recruitment and participation of racial and ethnic minorities in precision oncology

#### Internalized racism

*Internalized racism* encompassed two types of barriers namely: (i) *trust issues* and (ii) *social alienation.* We considered *trust issues* as being a manifestation of internalized racism. Past historical abusive events in the medical and research domains have been collectively internalized by minorities (i.e., their self-worth and dignity have been deemed inferior to Whites; Jones, 2000). In the empirical studies, we conceptualized *trust issues* of minorities as being related to: (i) the risks of their genetic and nongenetic information being misused, for instance, participants' genetic data being used without their consent or as a discriminatory feature against them (e.g., excluding some minority groups from accessing certain treatment options). In addition, some minorities feared that their donated biospecimens could be exploited commercially (sometimes referring to the case of Henrietta Lacks as an example; Diaz et al., 2014; Lee et al., 2019; Yeh et al., 2020); (ii) the risks of being manipulated as *guinea pigs* in research or medical care, which were often primed by past events (e.g., the Tuskegee Syphilis experiment; Lee et al., 2019; Loree et al., 2019; Yeh et al., 2020); (iii) general mistrust highlighting the importance of trusting healthcare providers (e.g., doctors and nurses) and researchers for engagement and retention in research (Keenan et al., 2015; Williams et al., 2018; Loree et al., 2019; Rencsok et al., 2020); and lastly (iv) privacy concerns, in particular concerning genetic data and biospecimens, that highlighted the need for 'develop[ing]

effective approaches toward building trust […] in precision medicine research' (Lee et al., 2019; Yeh et al., 2020).

*Social alienation*, it consisted of two elements, namely: (i) *perceived racism* and (ii) *social stigmatization. Perceived racism* was depicted only in the theoretical publications as an additional health stressor that worsened health outcomes for minorities (e.g., allostatic burden; Matthew, 2019; Geneviève et al., 2020; Zavala et al., 2021). *Social stigmatization* referred to the idea that the perceived social worth of minorities (e.g., their associated identities such as ethnicity and socioeconomic status) would likely dictate the availability of opportunities for them to access precision medicine interventions (Joly and Knoppers, 2014; Yeh et al., 2020).

#### Interpersonal racism

*Interpersonal racism* potentially accounted for four types of barriers to participation in precision oncology, namely: (i) *poor doctor/ genetic counselor–patient communication*, (ii) *implicit racial or ethnic bias*, (iii) *disrespect shown by healthcare professionals toward minority groups*, and (iv) minorities experiencing *higher delay between diagnosis and treatment, often followed by increased treatment discontinuation.* For the first barrier, Jagsi and colleagues noted that – even after controlling for other factors – '[m]inority patients were significantly more likely to have unmet need for discussion in this context [genetic testing for breast cancer], and those with this unmet need were much more likely to express worry about breast cancer as long-term survivors' (Jagsi et al., 2015). In a similar vein, the study by Cragun et al. investigating racial disparities in BRCA testing also showed that in comparison with non-Hispanic Whites, 'Blacks were 16.6 times less likely to have discussed genetic testing with a healthcare provider ($P < .0001$) […] after controlling for other variables' (Cragun et al., 2017). Since other factors have been accounted for, these findings suggest that

**Table 2.** Characteristics of included publications (*N* = 32)

| Reference | Article type | Location(s) (if applicable) | Study objective(s)* | Minority racial/ethnic group(s) | Cancer type(s) |
|---|---|---|---|---|---|
| Asmann et al. (2021) | Quantitative study | Canada, Italy, Spain, and United States | This study investigated how underrepresentation of minority groups in public germline variant databases could inflate tumor mutational burden in these groups. | Blacks | Multiple myeloma |
| Cragun et al. (2015) | Quantitative study | United States | This study evaluated the prevalence of referral and access to genetic counseling and genetic services, including their associated factors. | Blacks | Breast cancer |
| Cragun et al. (2017) | Quantitative study | United States | This study investigated and compared among BRCA [BReast CAncer gene] carriers their cancer-risk management practices, including discussion with healthcare providers concerning genetic testing and the actual receipt of the latter. | Blacks and Hispanics | Breast and ovarian cancers (BRCA-related) |
| Dai et al. (2022) | Quantitative study | China (where the authors are from) | This study used the TCGA (U.S.-based) database to investigate how race imbalance could influence the machine learning and statistical analyses for causal gene discoveries and patient survivorship predictions. | Blacks/African Americans, Asians, American Indians/Alaska Natives, Native Hawaiian/Other Pacific Islander, unknown | 32 cancer types |
| Diaz et al. (2014) | Quantitative study | United States | This study has evaluated variables that could influence and explain racial differences in attitudes for precision medicine, in particular self-reported discrimination and awareness. | Non-Hispanic Blacks | N/A |
| Jagsi et al. (2015) | Quantitative study | United States | Using population-based registries, this study identified and evaluated the experiences and preferences of a diverse group of breast cancer patients with regard to genetic testing while highlighting ethnoracial differences. | Blacks, Latinos, Other | Breast cancer |
| Keenan et al. (2015) | Quantitative study | United States | This study compared how breast cancer recurrence and tumor genomic characteristics were distributed between African American and white women. | African Americans | Breast cancer |
| Kehl et al. (2019) | Quantitative study | United States | This study evaluated disparities according to race and poverty in the initial uptake of molecular testing in Stage IV lung adenocarcinoma and to understand their importance by measuring the population-level association between biomarker-directed therapy and overall survival. | Black, Asian/other, Hispanic | Lung cancer |
| Landry et al. (2018) | Quantitative study | United States | A comparison of the numbers of studies in the GWAS Catalog, as well as the numbers of high-throughput sequencing studies in the database of Genotypes and Phenotypes, by ancestral population and disease area. | African Americans/African, Native Americans, Hispanics/Latinos | N/A |
| Lee et al. (2019) | Qualitative study | United States | This study determined whether patients distinguish between biospecimens and electronic health records when considering research participation to inform research protections. | African Americans, Chinese, Hispanics/Latinos, South Asians | N/A |
| Loree et al. (2019) | Quantitative study | United States | This study evaluated the frequency of race reporting and proportional race representation in trials supporting U.S. Food and Drug Administration oncology drug approvals. | Asians, Blacks, Hispanics | Solid and Hematological cancers |
| Myers et al. (2021) | Quantitative study | United States | This study investigated how perspectives on precision medicine may vary based on backgrounds (e.g., genomic vs. social behavioral) and research experience. | Asians, Blacks/African Americans, Hispanics/Latinos, American Indians/ Alaska Natives, Mixed | N/A |

*(Continued)*

**Table 2.** (*Continued*)

| Reference | Article type | Location(s) (if applicable) | Study objective(s)* | Minority racial/ethnic group(s) | Cancer type(s) |
|---|---|---|---|---|---|
| O'Neill et al. (2021) | Quantitative study | United States | This study investigated the association between cancer outcomes in African Americans and European Americans and p16 status in human papilloma virus-positive oropharyngeal squamous cell carcinoma. | African Americans | Oropharyngeal squamous cell carcinomas |
| Rencsok et al. (2020) | Quantitative study | Global | This study reviewed the reporting of race and ethnicity data and the representation of race and ethnicity across prostate cancer treatment-, prevention-, and screening-based clinical trials. | Blacks/African Americans, Asians, American Indians/Alaska natives, Native Hawaiians/other pacific islanders, Hispanics/Latinos, More than one race | Prostate cancer |
| Williams et al. (2018) | Quantitative study | United States | This study reported results from an analysis of survey data drawn from a diverse population in an effort to gain a better understanding of patient-level factors (ethnic/racial group and health literacy level) that may impede uptake and whether these patient-level factors influence perceptions across three precision medicine domains (genetics, behavior, and environment). | African Americans/Blacks, Hispanics/Latinos | N/A |
| Yeh et al. (2020) | Qualitative study | United States | This study aimed to better understand African-American and Hispanic perspectives on the potential benefits and barriers to reaping the benefits of precision medicine, including an assessment of differences in perspectives between these two groups. | African Americans, Hispanics | N/A |
| Zhu et al. (2016) | Quantitative study | United States | This study evaluated the performance of the preferential linkage disequilibrium approach in an African-American population, following up breast cancer GWAS hits. | African Americans | Breast cancer |
| Borrell et al. (2021) | Theoretical paper | N/A | This paper discussed the limitations and usefulness of self-identified race/ethnicity for precision medicine approaches. | N/A | N/A |
| Cohn et al. (2017) | Theoretical paper | N/A | This paper discussed distributive justice, diversity and inclusion in precision medicine. | N/A | N/A |
| Daly and Olopade (2015) | Theoretical paper | United States | This paper reviewed differences in the natural history, biology, genomics, and patterns of care of breast cancer in African Americans to understand racial disparity in mortality rates while reviewing innovative interventions to close the disparity gap. | Blacks/African Americans | Breast cancer |
| Carethers (2018) | Theoretical paper | United States | This paper explored some specific additional modifiable and nonmodifiable risk factors that may inform approaches to reduce the colorectal cancer disparity among African Americans. | African Americans | Colorectal cancer |
| Geneviève et al. (2020) | Theoretical paper | N/A | This study discussed the ways in which the implementation of precision medicine can be impacted by structural racism in the healthcare and research domains. | N/A | N/A |
| Joly and Knoppers (2014) | Theoretical paper | N/A | This study discussed challenges and solutions for an effective and equitable implementation of personalized medicine. | N/A | N/A |
| Kahn (2016) | Theoretical paper | Europe/ United States | This paper reviewed the racialization of patents for precision medicine applications. | N/A | N/A |

(*Continued*)

**Table 2.** (*Continued*)

| Reference | Article type | Location(s) (if applicable) | Study objective(s)* | Minority racial/ethnic group(s) | Cancer type(s) |
|---|---|---|---|---|---|
| Lim et al. (2014) | Theoretical paper | United States | This review mapped the current knowledge on the incidence and treatment outcome of childhood acute lymphoblastic leukemia in different populations, discussed the contribution of nongenetic and genetic factors while proposing some therapeutic interventions to reduce disparities along racial and ethnic lines. | Discussed in terms of ancestry rather than race or ethnicity: African, East Asian, and Native American | Childhood acute lymphoblastic leukemia |
| Matthew (2019) | Theoretical paper | N/A | This study discussed two challenges to precision medicine, namely scientific, and social racism while providing some recommendations to tackle them. | N/A | N/A |
| Mittendorf et al. (2021) | Theoretical paper | United States | This review provided insights on factors influencing the uptake and adherence of individuals suffering from hereditary cancer syndromes to guideline-recommended interventions aimed at reducing risks. | N/A | N/A |
| Rebbeck (2017) | Theoretical paper | United States | This paper explored the evidence for a role of germline genomics in explaining how prostate cancer risk, aggressiveness, and prognosis vary by race, ethnicity, and geography. | N/A | Prostate cancer |
| Spratt and Osborne (2015) | Theoretical paper | Europe/ United States | This paper discussed disparities in castration-resistant prostate cancer trials. | N/A | Prostate cancer |
| Thrall et al. (2021) | Theoretical paper | N/A | This study discussed the limits of applying a generalizable (one-size-fits-all) AI approach to radiological cancer screening and diagnostics, and that developing those specific to subpopulations (aligned with the precision medicine paradigm) is more appropriate. | N/A | Breast cancer |
| Underhill et al. (2016) | Theoretical paper | N/A | This study discussed the issues associated with cancer genetics risk assessment and testing for racial and ethnic disparities. | N/A | Hereditary breast and ovarian cancer |
| Zavala et al. (2021) | Theoretical paper | United States | This review summarized the reported disparities and associated factors in the United States for the most common cancers (breast, prostate, lung, and colon), and for a subset of other cancers that highlight the complexity of disparities (gastric, liver, pancreas, and leukemia). | African Americans/Blacks, American Indians and Alaska Natives, Asians, Native Hawaiians/other Pacific Islanders, Hispanics/Latinos | Breast, prostate, colorectal, and lung (+ other cancer types) |

*Abbreviation*: TCGA: The Cancer Genome Atlas. *For many papers, the objectives stated in Table 2 are as they appear in the original manuscripts.

the racial identity of participants has negatively impacted the quality of communication with healthcare providers. Therefore, the role of interpersonal racism – driven by either conscious or unconscious motivations – cannot be excluded. This leads us to the second barrier, *implicit racial or ethnic bias*, which were only reported in the theoretical publications. They were described mostly as barriers hindering access to high-quality care.

Concerning the third barrier, that is, *disrespect shown by health-care professionals toward minority groups*, it was only mentioned in the theoretical publications, whereby minorities often felt that their healthcare needs and expectations were not given due consideration by healthcare professionals. For the fourth barrier, it was mentioned in one empirical study and one theoretical study (Daly and Olopade, 2015; Keenan et al., 2015). Treatment and

screening delays were extensively covered by Daly and Olopade, and were reported to worsen survival rates in racial and ethnic minorities. In addition, treatment discontinuation/delays were also found to be more prevalent in minority groups than Whites, even if treatment-related toxicities have been taken into consideration (Daly and Olopade, 2015).

### Institutional racism
*Racial discrimination (perceived or real) from pharmaceutical companies and health insurers* was reported or identified as a barrier in numerous publications (Diaz et al., 2014; Joly and Knoppers, 2014; Underhill et al., 2016; Williams et al., 2018; Kehl et al., 2019; Lee et al., 2019; Matthew, 2019; Geneviève et al., 2020; Yeh et al., 2020; Borrell et al., 2021). For instance, a study comparing the attitudes of

**Table 3.** Types of racism and their associated barrier clusters

| Type of racism | Cluster |
| --- | --- |
| Internalized racism | Trust issues. |
| | Social alienation as perceived racism and social stigmatization. |
| Interpersonal racism | Poor doctor/genetic counselor – patient communication between racial minorities and healthcare professionals from the dominant group, even when other factors have been accounted for. |
| | Implicit racial and ethnic biases. |
| | Disrespect from healthcare staff/researchers to racial and ethnic minorities. |
| | Higher delay between diagnosis and treatment in comparison to the majority racial/ethnic group, often leading to increased treatment discontinuation for racial/ethnic minorities. |
| Institutional racism | Racial discrimination from pharmaceutical industries/health insurers. |
| | Failing healthcare institutions/research enterprises for minorities. |
| | *New racism* (DiAngelo, 2012), whereby exclusion of minorities is perpetuated implicitly and often rationally by using cultural differences (e.g., language spoken) – rather than racial identities – as discriminatory features in the allocation of resources. |
| | Institutional blindness to racism (e.g., research and pharmaceutical enterprises might implicitly reiterate scientific racism or fail to factor in racism as a potential contributor to observed cancer disparities between racial and ethnic groups). |
| Structural racism | Underrepresentation of racial/ethnic minorities as patients/study participants, and its consequences/causes. |
| | Environmental racism. |

**Table 4.** Reporting of barriers ($N = 135$) per cluster and per study type

| | Empirical studies ($n^a = 58$) | Theoretical publications ($n^a = 77$) |
| --- | --- | --- |
| Internalized racism (no. of barriers, %[b]) | • Trust issues (13, 22.4%) (Diaz et al., 2014; Keenan et al., 2015; Williams et al., 2018; Lee et al., 2019; Loree et al., 2019; Rencsok et al., 2020; Yeh et al., 2020)<br>• Social alienation (1, 1.7%) (Yeh et al., 2020) | • Trust issues (9, 11.7%) (Daly and Olopade, 2015; Underhill et al., 2016; Cohn et al., 2017; Matthew, 2019; Geneviève et al., 2020; Mittendorf et al., 2021)<br>• Social alienation (4, 5.2%) (Joly and Knoppers, 2014; Matthew, 2019; Geneviève et al., 2020; Zavala et al., 2021) |
| Interpersonal racism (no. of barriers, %) | • Poor doctor/genetic counselor–patient communication (2, 3.4%) (Jagsi et al., 2015; Cragun et al., 2017)<br>• Higher delay leading to +/− increased treatment discontinuation (1, 1.7%) (Keenan et al., 2015) | • Poor doctor/genetic counselor–patient communication (2, 2.6%) (Daly and Olopade, 2015; Geneviève et al., 2020)<br>• Implicit racial or ethnic bias (3, 3.9%) (Daly and Olopade, 2015; Geneviève et al., 2020; Borrell et al., 2021)<br>• Disrespect from healthcare staff/researchers (2, 2.6%) (Daly and Olopade, 2015; Geneviève et al., 2020)<br>• Higher delay leading to +/− increased treatment discontinuation (1, 1.3%) (Daly and Olopade, 2015) |
| Institutional racism (no. of barriers, %) | • Racial discrimination from pharmaceutical industries/health insurers (5, 8.6%) (Diaz et al., 2014; Williams et al., 2018; Kehl et al., 2019; Lee et al., 2019; Yeh et al., 2020)<br>• Failing healthcare institutions/research enterprises (10, 17.2%) (Diaz et al., 2014; Cragun et al., 2015; Jagsi et al., 2015; Keenan et al., 2015; Cragun et al., 2017; Landry et al., 2018; Williams et al., 2018; Kehl et al., 2019; Loree et al., 2019; Yeh et al., 2020)<br>• New racism (2, 3.4%) (Landry et al., 2018; Loree et al., 2019)<br>• Institutional blindness to racism (1, 1.7%) (Myers et al., 2021) | • Racial discrimination from pharmaceutical industries/health insurers (5, 6.5%) (Joly and Knoppers, 2014; Underhill et al., 2016; Matthew, 2019; Geneviève et al., 2020; Borrell et al., 2021)<br>• Failing healthcare institutions/research enterprises (14, 18.2%) (Joly and Knoppers, 2014; Lim et al., 2014; Daly and Olopade, 2015; Spratt and Osborne, 2015; Underhill et al., 2016; Matthew, 2019; Geneviève et al., 2020; Borrell et al., 2021; Mittendorf et al., 2021; Zavala et al., 2021)<br>• New racism (2, 2.6%) (Daly and Olopade, 2015; Zavala et al., 2021)<br>• Institutional blindness to racism (5, 6.5%) (Kahn, 2016; Cohn et al., 2017; Matthew, 2019; Zavala et al., 2021) |
| Structural racism (no. of barriers, %) | • Underrepresentation of racial and ethnic minorities (8, 13.7%) (Zhu et al., 2016; Landry et al., 2018; Lee et al., 2019; Loree et al., 2019; Rencsok et al., 2020; Asmann et al., 2021; O'Neill et al., 2021; Dai et al., 2022) and its causes (10, 17.2%) (Cragun et al., 2015; Jagsi et al., 2015; Cragun et al., 2017; Williams et al., 2018; Loree et al., 2019; Rencsok et al., 2020) and consequences (4, 6.9%) (Zhu et al., 2016; Dai et al., 2022)<br>• Environmental racism (1, 1.7%) (Keenan et al., 2015) | • Underrepresentation of racial and ethnic minorities (10, 13.0%) (Daly and Olopade, 2015; Spratt and Osborne, 2015; Underhill et al., 2016; Cohn et al., 2017; Rebbeck, 2017; Carethers, 2018; Matthew, 2019; Geneviève et al., 2020; Thrall et al., 2021; Zavala et al., 2021) and its causes (12, 15.6%) (Daly and Olopade, 2015; Underhill et al., 2016; Matthew, 2019; Geneviève et al., 2020; Borrell et al., 2021; Zavala et al., 2021) and consequences (8, 10.4%) (Daly and Olopade, 2015; Rebbeck, 2017; Geneviève et al., 2020; Borrell et al., 2021; Thrall et al., 2021; Zavala et al., 2021) |

[a]$n$ = Total number of barriers per publication type (empirical or theoretical).
[b]Percentage of barriers per publication type (empirical or theoretical).

non-Hispanic Whites and Blacks toward personalized medicine found that – in general – approximately 75% of study participants were reluctant to share their genetic information with health insurers due to the fear of being discriminated (Diaz et al., 2014). Similarly, in another study, minority participants viewed insurance companies as being more problematic than healthcare providers concerning access to precision medicine interventions (Yeh et al., 2020). Regarding the risk of racial discrimination in drug development, Geneviève and colleagues argued that pharmaceutical companies might implicitly or explicitly (if motivated by financial gains) favor the development of new therapeutics for diseases afflicting more Whites to the detriment of others (Geneviève et al., 2020).

Concerning *failing healthcare institutions/research enterprises for racial and ethnic minorities*, this barrier consisted of three distinctive components that render the recruitment and participation of racial and ethnic minorities within the precision medicine paradigm problematic. The first component concerns how healthcare institutions have been designed in general to answer the needs of the majority group. Therefore, the ability of minorities to access healthcare services is already limited (Geneviève et al., 2020; Zavala et al., 2021). The second component concerns how cancer care tends to be of lower quality or less definitive for minorities. For instance, it has been observed that African-American men suffering from aggressive or high-risk prostate cancer are less likely to be offered definitive surgical or radiotherapy treatment – although recommended – than White patients (Zavala et al., 2021). Similarly, another study also noted that cancer treatment tends to be underused or misused for racial and ethnic minority groups, which could partially explain why disparities in cancer survival rates persist (Daly and Olopade, 2015). The third component concerns how access to cancer screening procedures, follow-up care, and clinical trials can be limited by a number of different but sometimes interconnected factors such as: (i) the absence of available therapeutic options for racial and ethnic minorities and (ii) the absence of referral by healthcare professionals of minority patients to specialized screening and therapeutic interventions despite clear modalities being established in cancer care protocols (Diaz et al., 2014; Joly and Knoppers, 2014; Cragun et al., 2015; Jagsi et al., 2015; Keenan et al., 2015; Spratt and Osborne, 2015; Underhill et al., 2016; Landry et al., 2018; Williams et al., 2018; Kehl et al., 2019; Loree et al., 2019; Matthew, 2019; Geneviève et al., 2020; Yeh et al., 2020; Borrell et al., 2021; Zavala et al., 2021).

With regard to the *cultural limitations of institutions as a form of new racism*, the current recruitment strategies for participation in cancer clinical trials have been limited in their effectiveness to recruit racial and ethnic minority populations. Just like precision oncology goes against a one-size-fits-all approach to individualize and maximize the effectiveness of cancer care interventions, recruitment strategies are likely to be ineffective in enhancing the diversity of clinical trials if they adopt a general approach that ignores the cultural background of participants (Landry et al., 2018; Loree et al., 2019). The same limitations also apply to healthcare institutions and insurers – in particular when care needs to be coordinated by minorities themselves between these two institutions – where the lack of culturally (and linguistically) tailored programs and follow-up strategies have been reported as a barrier to the participation of racial and ethnic minorities (Daly and Olopade, 2015; Zavala et al., 2021).

The last institutional barrier concerns how *institutional blindness to racism* can lead research and pharmaceutical enterprises to implicitly reiterate *scientific racism* or ignore the contribution of racism to observed cancer disparities (Kahn, 2016; Cohn et al., 2017;

Matthew, 2019; Myers et al., 2021; Zavala et al., 2021). For instance, Matthew explicitly mentioned *scientific racism* as a health equity threat to precision medicine initiatives. Indeed, scientific racism risks misdirecting the focus of precision medicine interventions toward attributing racial disparities to innate biological or genetic differences between the so-called races rather than meaningfully investigating and eliminating these health inequities (e.g., by considering the often ignored social determinants of health; Matthew, 2019). In addition, scientific racism can also emerge from current inclusion metrics. Indeed, the Office of Migration and Budget's self-reported racial categories have been privileged in genomic studies. Incorporating these racial categories in genomic studies could reiterate the notion of biological races as a potential explanation for health disparities (Cohn et al., 2017). Furthermore, with the advent of precision medicine, racial categories are increasingly being geneticized in biomedical practice and drug patenting, with the risks that some racial groups could be deprived from accessing a certain medication or treatment option although they are not significantly different on a genetic or physiological basis from other racial groups (Kahn, 2016).

On another note, it has been reported that researchers often neglect to examine the relationship between racism and genomic differences between racialized groups, and how this complex interaction could explain some cancer disparities (Zavala et al., 2021). An interesting insight brought forward by Myers et al. concerned how different disciplines involved in precision medicine activities influence the perception of individuals working therein (genetic researchers vs. sociobehavioral scientists). Indeed, those involved in genetic research were found to be less likely to recognize the effect that racism and discrimination could have on the results of genetic testing than those involved in the sociobehavioral sphere. This finding highlighted the need for cross-disciplinary training (Myers et al., 2021).

### Structural racism

Structural racism accounted for two types of barriers to recruitment and participation, namely: (i) the *underrepresentation of racial and ethnic minority groups* in precision oncology activities (including the *causes* and *consequences* of such underrepresentation for inclusion and participation) and (ii) *environmental racism.* The *underrepresentation of racial and ethnic minority groups* in genomic databases, clinical trials, biobanks, and other elements necessary for the proper functioning of precision oncology activities has been reported/identified in 19 out of 32 studies included in this scoping review (Daly and Olopade, 2015; Spratt and Osborne, 2015; Underhill et al., 2016; Zhu et al., 2016; Cohn et al., 2017; Rebbeck, 2017; Carethers, 2018; Landry et al., 2018; Lee et al., 2019; Loree et al., 2019; Matthew, 2019; Geneviève et al., 2020; Rencsok et al., 2020; Asmann et al., 2021; Borrell et al., 2021; O'Neill et al., 2021; Thrall et al., 2021; Zavala et al., 2021; Dai et al., 2022).

Some of the identified potential *causes of the underrepresentation* of minorities include: (i) residential segregation and low socioeconomic neighborhoods (Matthew, 2019; Geneviève et al., 2020; Yeh et al., 2020; Zavala et al., 2021); (ii) the relatively low educational status of minorities (Cragun et al., 2015; Cragun et al., 2017); (iii) low participant–researcher or patient–provider racial and ethnic concordance (Loree et al., 2019; Matthew, 2019; Geneviève et al., 2020); (iv) lack of information on or awareness of precision oncology (treatment or clinical trials) and related activities (Cragun et al., 2015; Daly and Olopade, 2015; Jagsi et al., 2015; Underhill et al., 2016; Cragun et al., 2017; Williams et al., 2018; Matthew, 2019; Geneviève et al., 2020; Rencsok et al., 2020; Zavala et al., 2021); (v) lack of resources to carry out research or treatment with racial and ethnic minorities (Daly and Olopade, 2015; Zavala et al., 2021);

and in a similar vein, (vi) private funding being available mostly for research aimed to be conducted with White participants (Rencsok et al., 2020).

The *consequences of this underrepresentation* are: (i) the risks of *algorithmic discrimination*, in particular if artificial intelligence tools have been trained on biased datasets (Geneviève et al., 2020; Thrall et al., 2021; Dai et al., 2022). For instance, in AI-enabled genomic analyses, Dai et al. demonstrated, via TCGA database, that a high degree of racial bias reduces not only the discovery of cancer causal genes, but also the accuracy of survival prediction models for minority populations (Dai et al., 2022); (ii) limited use and effectiveness of polygenic risk scores for predicting cancer risk in racial and ethnic minorities (Borrell et al., 2021; Zavala et al., 2021); (iii) inadequate risk stratification biomarkers or clinical risk models (Zavala et al., 2021; Dai et al., 2022); (iv) limited knowledge on cancer biomarkers that are specific to racial and ethnic minorities (Rebbeck, 2017); (v) lack of research on genetic abnormalities in racial and ethnic minorities, which subsequently contribute to a higher prevalence of genetic variants of unknown significance in these groups (e.g., in African Americans; Daly and Olopade, 2015; Zhu et al., 2016; Rebbeck, 2017); and (vi) results of GWASs relying predominantly on populations of European descent have limited applicability to minority groups (Zhu et al., 2016).

Only one empirical study conducted by Keenan and colleagues (Keenan et al., 2015) may have provided evidence on the contribution of *environmental racism* to breast cancer disparities between White and African-American women. Their results showed that breast cancer of African-American women had a higher intratumor genetic heterogeneity than White women. One potential explanation to this discrepancy is that African-American women have a higher exposure to environmental mutagens because of their lower socioeconomic conditions (due to structural racism). The higher intratumor genetic heterogeneity implies that their tumors are likely to be more resistant to current and new treatment options, in particular for therapies directed at specific genotypes (Keenan et al., 2015).

## Application of PHCRP to the design and findings of empirical studies

Within the PHCRP framework, four factors became evident that influence not only the recruitment and participation of minorities in precision oncology activities, but also how the knowledge generated from these studies could be curtailed by assumptions or simplifications made during the research process (they are explained below).

### Lack of conceptual clarity on the meaning and use of race and ethnicity

The conceptual boundaries between the distinct concepts of race and ethnicity were shown to be ill-defined in the included empirical studies. These distinct concepts were often used and treated by researchers as synonyms or taken together as one specific variable (Diaz et al., 2014; Jagsi et al., 2015; Williams et al., 2018; Loree et al., 2019; Rencsok et al., 2020; Asmann et al., 2021; Myers et al., 2021). Sometimes, ethnicities and racial identities were analyzed together as one variable for statistical considerations due to either low numbers of specific minority group participants (Diaz et al., 2014; Landry et al., 2018; Williams et al., 2018) or for statistical convenience (Loree et al., 2019; Rencsok et al., 2020).

### Monolithic view of racial identities

Racial identities – self-reported or administrative/assumed race – were often viewed and treated as monolithic categories in empirical studies investigating genetic or biological associations with certain risk factors or health outcomes (Keenan et al., 2015; Asmann et al., 2021; O'Neill et al., 2021). For instance, in the study by Asmann and colleagues (Asmann et al., 2021), the authors have not only acknowledged the limitations posed by self-reported race for patient selection, but also that the concept of *race* is socially constructed. Yet, the authors had no other choice in their study but to treat racial categories as monoliths since this is how these groups are categorized in publicly available databases (Asmann et al., 2021). Landry and colleagues made a similar observation in their study, stating that '[o]ne of the most striking findings [in their study] is the lack of ancestral information included in these datasets, specifically in the database of Genotypes and Phenotypes' (Landry et al., 2018). This monolithic view becomes even more problematic when 'race' is being considered as a good biological proxy, for instance, in the study by O'Neill and colleagues (O'Neill et al., 2021). Indeed, the authors argued that 'HR [hazard ratio] of race was, *impressively* [emphasis added], similar to p16 status, further demonstrating the high clinical impact of race as a prognostic biomarker in OPSCC [oropharyngeal squamous cell carcinoma], independent of p16 status' (O'Neill et al., 2021). This can lead to neglect the role played by the social determinants of health (including racism) or simply that spurious correlations could have been responsible for the observed disparities in prognosis.

### The Black–White binary paradigm

The *Black–White binary paradigm* can be defined as 'the conception that race in America consists, either exclusively or primarily, of only two constituent racial groups, the Black and the White […] only the Black and the White races matter for the purposes of discussing race and social policy with regard to race' (Perea, 1997). The pervasiveness of this paradigm in the United States was reflected in the empirical studies, whereby 6 out of the 17 empirical studies were focused exclusively on investigating issues affecting African Americans/Blacks or obtaining their perspectives on certain aspects of genetic testing, precision medicine, or precision oncology (Diaz et al., 2014; Cragun et al., 2015; Keenan et al., 2015; Zhu et al., 2016; Asmann et al., 2021; O'Neill et al., 2021). Furthermore, it is argued that 'the paradigm dictates that all other racial identities and groups in the United States are best understood through the Black/White binary paradigm' (Perea, 1997). Indeed, African Americans are commonly viewed as the prototypical minority group (Delgado and Stefancic, 2017a), that is, the worse-off group. Therefore, study results based on African-American participants are often presumed to be relevant to other minority groups. For instance, in one of the studies involving African Americans, it is stated that 'the inflated TMBs [Tumor Mutational Burdens] are likely relevant for other ethnic groups including Asians, Pacific Islanders, and other underrepresented groups' (Asmann et al., 2021). In addition, the Black–White binary paradigm not only seems to prioritize African Americans for health disparities research, but also shapes certain parameters of multiracial studies (i.e., studies involving other minority groups). For instance, the objective of the study conducted by Rencsok and colleagues was to investigate and assess the reporting of the ethnicity and race of study participants involved in prostate cancer clinical trials, including their representation. In that regard, the authors proposed some participation benchmarks

for the inclusion of non-Hispanic Whites and non-Hispanic Blacks in prostate cancer clinical trials, but failed to do so for other racial minority groups. Nonetheless, this limitation was acknowledged by the authors: '[we, the authors] specifically mention non-Hispanic White and African-American men in our proposed benchmarks; regardless, the inclusion of U.S. minority groups other than African Americans should also be prioritized' (Rencsok et al., 2020).

### Not naming and ignoring the contribution of racism to observed disparities

Racism was explicitly mentioned in only 2 of the 17 empirical studies (Yeh et al., 2020; Myers et al., 2021; 11.7%; these may be due to the study methodologies involved: survey and qualitative research). For instance, in the study by Williams and colleagues on perceived drivers of personal health in African Americans and Hispanics, African Americans reported that their race and ethnicity and the resulting discrimination played a role in their health. However, rather than naming 'racism' as a potential contributor to observed disparities, the authors preferred to acknowledge the role of cultural experiences in the observed perceptions in precision medicine (Williams et al., 2018).

### Discussion

To the best of our knowledge, this is the first scoping review mapping the recruitment and participation barriers that racial and ethnic minority populations are likely to encounter in precision oncology and other related activities (e.g., cancer genomic studies). Our findings showed that the majority of barriers identified or reported were linked to structural racism, followed by institutional, internalized, and lastly, interpersonal forms of racism. In addition, our critical analysis of the empirical studies identified four factors that are likely to constitute additional barriers to the equitable inclusion and diversity goals of precision oncology. These barriers and factors are discussed in the following sections.

Trust issues were the predominant barrier for internalized racism, often primed by historical events where minorities were abused in science or in medical care. The resulting and long-lasting mistrust felt by minorities – although legitimate – is possibly contributing to the worsening of health disparities by preventing or limiting access to newer and potentially more effective cancer screening and therapies derived from precision oncology. Therefore, the importance of favoring trusting relationships between minorities and stakeholders involved in precision oncology studies (e.g., via community engagement to better understand their needs and fears with regard to research participation) cannot be underestimated, as highlighted by many initiatives working toward increasing diversity in cancer clinical trials (Regnante et al., 2019; Oyer et al., 2022). In this regard, one potential trust-enhancing partnership model for precision oncology could be a community-based participatory research one, which has so far been successful in health disparities research. This model ensures that project leadership is equitably shared during all phases of the research project between the community and the research team, and it relies on 'principles of trust, respect, power sharing, and transparency through two-way knowledge exchange and finding the "win–win" to maintain engagement' (Israel Barbara et al., 2005; Jones et al., 2018). It is also important to understand that past historical abuses in the medical and research domains are still present in racial and ethnic minorities'

mind and influence their behavior toward the healthcare system and biomedical research. Therefore, it is also crucial: (i) to disclose in a transparent and intelligible manner the study results, and the expected benefits of research participation for minorities (and potentially future generations), (ii) to have a trusted recruiter who minorities can relate to in terms of similar research experience and identity (Hughes et al., 2017), or at least have someone from the research team being introduced to the community via a community liaison (Mancera et al., 2021), and lastly (iii) to promote the sharing of counter narratives of past and current successful minority participation in research.

With reference to interpersonal racism, implicit racial and ethnic biases were among the main barriers. These biases have been extensively covered in the scientific literature (see reviews Hall et al. (2015) and FitzGerald and Hurst (2017)) and were found to potentially contribute to health disparities. They may become even more problematic in the era of precision oncology, whereby a data deluge (with the addition of new molecular variables) will have to be analyzed and interpreted by healthcare professionals to support clinical decision-making. However, human cognitive capabilities are likely to be exceeded in precision oncology, in particular if effective decision support tools are not available (Johnson et al., 2016; Walsh et al., 2019). Therefore, reliance on heuristics and implicit racial biases to support decision-making will likely increase due to cognitive overload (Johnson et al., 2016). Furthermore, these implicit biases may also be contributing to other barriers associated with interpersonal racism, such as communication issues with healthcare providers (e.g., genetic counselors; Schaa et al., 2015), or even healthcare professionals disrespecting or initiating treatment at a relatively later stage for minorities. Such negative experiences are likely to further substantiate their feelings of mistrust and hamper progress that is direly needed to ensure equal access to the benefits that precision oncology could provide. Despite their potential negative contribution to observed cancer disparities, implicit racial and ethnic biases were only reported in the theoretical publications. The lack of empirical data on implicit racial and ethnic biases may reside in the methodological difficulties to measure reliably these biases (Maina et al., 2018), or due to the fact that some researchers (as argued in Myers et al. (2021) and in the PHCRP analysis) tend to abide to the illusion that their respective fields are objective and free from racist influences. Regarding the latter, we believe that there is a need for genetic researchers and oncologists to undergo cross-disciplinary training with social and behavioral scientists (Myers et al., 2021), since the proper management of cancer patients should go beyond purely medical considerations to also include sociocultural ones such as racism (Kagawa-Singer, 2000).

The barriers related to institutional and structural racism were the most frequently reported/identified in the manuscripts included in this scoping review. Addressing these barriers is of paramount importance not only for the success and sustainability of precision oncology initiatives, but also for addressing the long-standing cancer disparities gap afflicting minorities. In that regard, one central barrier to address concerns the underrepresentation of minorities, which lies at a critical nexus between new scientific findings emerging from precision oncology and genomic studies, and their often limited transferability to racial minorities (Rebbeck, 2017; Yeh et al., 2020; Borrell et al., 2021; Zavala et al., 2021; Dai et al., 2022). Our scoping review also showed that the reasons for the underrepresentation of minorities are multifactorial, and likely to permeate from structural racism to all other forms of racism. Therefore, we argue that interventions aimed at remedying the situation should primarily tackle the barriers associated with

institutional and structural racism, which should then have ripple effects on other forms of racism. Another important barrier is the cultural and linguistic limitations of institutions, which we consider to be another form of 'new racism' (DiAngelo, 2012). *New racism* can be defined as, '[t]he ways in which racism has adapted over time so that modern norms, policies, and practices result in similar racial outcomes as those in the past, while not appearing explicitly racist' (DiAngelo, 2012). Research and healthcare institutions may rightly argue – under the motive of utilitarianism (Stanford Encyclopedia of Philosophy, 2014) – that with their often limited resources, they cannot afford to stretch their services to include culturally competent programs for all minorities, but they have to offer services to the highest number of people (and hence the majority group is prioritized). However, if institutions are not trying to offer culturally and linguistically adapted services or research programs to all members of the society they are supposed to serve, they are *de facto* perpetuating for cultural minorities the systematic exclusion from research studies while contributing to worsened health outcomes. Therefore, it becomes the role of the researchers, clinicians, funders, and policy-makers to work in a concerted manner to implement measures to solve the limitations that researchers and clinicians claim as to why culture and linguistic issues cannot be solved.

Our PHCRP-guided analysis of the empirical studies also revealed the presence of four additional factors that could further explain the limited recruitment and participation of minorities in precision oncology. One of them concerns the lack of conceptual clarity on the meaning and use of race and ethnicity. In that regard, it is crucial for clinical geneticists and biomedical researchers to comprehend how, when, and when not to use these socially constructed concepts (i.e., human-invented categories that have no biological foundations but still ascribed racialized groups political, economic, and societal power and opportunities) in their daily practice or research projects. These socially constructed concepts shall not be used as indicators for the presence of certain genetic variants in racialized populations, but rather used to investigate how self-identified race can lead to systemic discrimination and bias in terms of access and participation to precision oncology (Rebbeck et al., 2022). However, this could be a challenging task for clinicians and researchers for at least two reasons: (i) the fluidity of these concepts implies that there are up-to-now no standardized definitions and evidence-based use of race and ethnicity and (ii) the limited availability of genetic ancestry information (either during clinical consultations or in genomic databases – see examples in Asmann et al. (2021) and Landry et al. (2018)) nudges researchers to resort to the use of race and ethnicities as proxies for genetic ancestry (Popejoy et al., 2020). Therefore, determining the presence of certain genetic variants shall only be carried out through genetic testing, and not presumed based on fluid and socially constructed identities.

In that regard, Popejoy and colleagues have rightfully argued on the need to have clear and standardized definitions (and evidence-based use) of race and ethnicity that would inform consistently the whole chain of knowledge generation and application in precision oncology, from the clinical consultation in the oncologist's office to laboratories, genomic and phenotypic databases, and research enterprises (Popejoy et al., 2020). Nevertheless, racial categories still carry the risk of being misinterpreted as biological constructs, and hence they can contribute implicitly to the sustainability and reinforcement of racism in the scientific and healthcare domains. Therefore, we could not agree more with the proposal made by Braveman and Dominguez to substitute U.S. racial categories with ethnic ones (which is common practice in many European countries). Indeed, the latter are not only more scientifically relevant

than 'race', given that racial categories do not take into account the considerable genetic admixture that currently exists within racialized groups, but ethnic categories can also be used – like racial categories – to monitor the influence of racism on minorities (Braveman and Parker Dominguez, 2021).

Another factor concerns the tendency of researchers and other stakeholders to view and group racialized populations as monoliths in research, thereby wrongly equating self-reported race of study participants to the presence of uniform genetic characteristics or specific genetic attributes. Importantly, our review also revealed that current tools used in genomic studies could implicitly nudge researchers to embrace further this monolithic view of racial identities. Some genomic databases are prototypical examples of how limitations in certain tools used for genetic research can implicitly influence researchers to reiterate scientific racism, whereby racial categories – although socially constructed – are perceived as being genetically homogeneous and therefore exhibiting innate biological differences. In addition, these tools fail to capture and acknowledge the genetic diversity that can exist within taxonomized racial groups. More widely, such a monolithic view can also undermine the achievement of true ancestral genetic diversity. Indeed, the majority of participants of African ancestry in GWASs are part of African diasporas, which themselves originate from only one of the five main African ethnolinguistic divisions (Fatumo et al., 2022). In this case, '[s]tudying a small number African diaspora populations […] and grouping all participants into a broad category of African ancestry will continue to promote imbalance, widen health disparities, and will fail to capture the genetic diversity in Africa' (Fatumo et al., 2022). Therefore, it is paramount to advocate for the collection of genetic ancestry data during clinical encounters (with the setting up of appropriate data safeguards and trust-enhancing measures to protect minorities), including the mandatory inclusion of genetic ancestral information in genomic databases to help provide a more accurate and fuller picture of 'disease-gene and -variant associations' (Popejoy et al., 2018; than relying exclusively on inaccurate proxies such as race). These measures will also help in reducing the risks of misinterpreting genetic tests (Jorde and Bamshad, 2020).

With regard to the *Black–White binary paradigm*, its implicit adoption by researchers in precision oncology can be problematic not only for the inclusion and participation of other non-Black minority groups, but also for the generation of knowledge to tackle cancer in these groups. Indeed, all paradigms in research define – in one way or another – the facts that are scientifically important and therefore deserving of investigation. However, they also ignore or exclude other facts or valid alternative theories that do not fit within the boundaries set by the latter paradigms (Perea, 1997). Therefore, knowledge on specific pathways leading to cancer disparities in non-Black minorities – which could be different from those affecting African Americans – is also likely to remain limited unless the binary paradigm is changed. Indeed, racial oppression can occur via different axes, depending on the racial minority group concerned. For instance, Latinos and Asian Americans are often oppressed via other axes than skin color, such as those concerned with cultural origin, nativism, or other physical features (Alcoff, 2005). In addition, the Black–White binary paradigm can weaken solidarity between racial minority groups in their fight against racial injustice (Delgado and Stefancic, 2017a). On top of hindering coalition between racial minorities, this dichotomous view of the races limits the sharing of useful knowledge generated from non-Black minority groups, that could have benefitted African Americans (Delgado and Stefancic, 2017a).

The last identified factor was that racism was rarely mentioned or acknowledged as a contributing factor to observed cancer disparities between ethno-racial groups. Unfortunately, ignoring the contribution of racism to health disparities is a common phenomenon in medical research. Back in 2018, a systematic review showed that *institutionalized racism* was often not explicitly mentioned in the titles or abstracts of public health studies in the United States, even if dealing with this specific type of racism (Hardeman et al., 2018). More recently, Krieger et al. also showed that leading gatekeeper medical journals tend not to publish credible research on racism and the few exceptionally published ones were mostly letters, viewpoints, or commentaries (Krieger et al., 2021). By not acknowledging explicitly the contribution of racism, researchers and healthcare professionals are not only depriving the antiracism scholarship from useful insights to monitor the evolution of structural racism occurring therein, but also contributing to the illusion that their respective fields are free from such influences. Therefore, we recommend that researchers active in precision oncology and editors/peer-reviewers of scientific journals to always consider the contribution of racism (and not only 'race' as a variable) as a potential explanation for observed cancer disparities in empirical studies. Research ethics committees could also play a more active role in evaluating health disparities studies by requiring researchers to adopt an antiracist approach when designing their respective projects.

## Limitations

This scoping review has some limitations. First, only two databases were systematically searched, and therefore it is likely that additional studies that could have provided additional insights were not included. Second, the scoping review methodology has some inherent limitations, whereby a focus is placed on the breadth of analysis to map the barriers reported in the literature, and therefore the quality of included records is not assessed. Third, it is possible that we may have missed important terms in our search strategy, resulting in missing relevant papers in the two databases. We, however, hope that the additional reference checking which results in many papers that we included would have addressed part of these limitations. Fourth, with regard to the PHCRP framework, we – as foreigners – have found it challenging to capture racialization processes that are specific to the United States, although B.S.E. has worked there for some time and T.W. completed her entire higher education (9 years) in the United States. In addition, the challenge with its application to the results of empirical studies is that the latter often lack the space necessary to provide a detailed account of events that could help to better understand how racialization processes are embedded in the research process (e.g., journal editorial policies that limit the word count of manuscripts).

## Conclusions

Being at the forefront of the precision medicine paradigm, precision oncology offers an avant-garde window into how structural racism can limit inclusion and diversity, and subsequently restricts the benefits that racialized minorities can reap from such advances in cancer care. Consequently, it is essential for researchers, policy-makers, and other stakeholders to be better prepared in tackling the insidious influences structural racism can have not only on their respective projects, but also on knowledge production. Concerning the latter and in the words of Mya Roberson: '[w]e cannot continue to herald research generated from populations primarily racialized as White as the highest standards of evidence, as it is not an accurate reflection of a global society' (Roberson, 2022). To this aim, this scoping review provides insights on some of the recruitment and participation barriers that racial and ethnic minorities are likely to face in precision oncology and other related activities. Moreover, it is crucial to understand that many barriers (if not all) – even if they have been linked to other forms of racism in this scoping review – are actually downstream manifestations of structural racism acting at the individual, interpersonal, and institutional levels.

It is also paramount for stakeholders active in precision oncology (e.g., oncologists, researchers, and owners of genomic databases) to also reflect and act on the four factors identified with the PHCRP analysis framework. For instance, an improved workforce diversity could help in tackling many of the identified barriers, and efforts in this direction are currently being made by the UNITE initiative. In addition, the U.S. NIH have developed a data dashboard, where inclusion and diversity aggregated data on different domains (e.g., on NIH workforce demographics and grantees) are made available to the public (National Institutes of Health, 2021). However, it is also important to advocate for meaningful representation of minorities and guard against tokenism. For example, the meaningful representation can be promoted by establishing racial and ethnic diversity thresholds at all hierarchy levels of institutions active in precision oncology. Moreover, these institutions also have the moral duty to publicly justify why certain minorities are underrepresented in their workforce and how they plan to remedy to the situation. Although such measures might seem drastic in their approach (and many will even question their feasibility or even their value), it is important to comprehend that against structural racism, positive changes to reverse its long-lasting damages and prejudices on racial minorities can only occur through radical measures (Delgado and Stefancic, 2017b). Without a radical approach to health equity, any action against structural racism will be futile, since the systemic discriminatory pathways will morph accordingly to maintain the *status quo*.

**Open peer review.** To view the open peer review materials for this article, please visit http://doi.org/10.1017/pcm.2022.4.

**Data availability statement.** The authors confirm that the data supporting the findings of this study are available within the article.

**Author contributions.** L.D.G. conceptualized the manuscript, performed the data charting and PHCRP analysis, wrote the original draft, and edited and reviewed subsequent versions of the manuscript following the suggestions and modifications made by other co-authors. B.S.E. was involved in the conceptualization of the manuscript, and edited and reviewed subsequent versions of the manuscript. T.W. was involved in the conceptualization of the manuscript, helped in the data analysis process, and edited and reviewed subsequent versions of the manuscript. The authors have read and agreed on the version of the manuscript to be published.

**Financial support.** L.D.G. acknowledges the financial support provided by the Käthe-Zingg-Schwichtenberg-Fonds of the Swiss Academy of Medical Sciences (Grant No. KZS 08/20). The funder played no role in the writing nor in the decision to publish this research work.

**Competing interest.** The authors declare no competing interests exist.

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
