## [Reviewer Report]

*Comments to Author*: This timely, insightful paper sets out a comprehensive analysis and evaluation of barriers arising from various types of racism to the recruitment and participation of minorities in precision oncology research. The authors reviewed empirical and theoretical literature from the Web of Science and PubMed, and then applied the Public Health Critical Race Praxis (PHCRP) as a guiding analytical framework to identify barriers arising during the research process. 135 barriers were identified, and the authors present a plausible and logical (and ubiquitous) list of barriers arising in the empirical and theoretical literature.

I would support the publication of the paper subject to two minor revisions:

• It would have been helpful to understand whether this analytical method was able to discriminate between a brief mention and a deep analysis in the papers quoted. [Table 4 reports the barriers in terms of total number of barriers per publication type and percentage of barriers per publication type]. The concern is that this method may have underestimated the impact of papers containing a rich but limited discussion.

• It is only towards the end of the paper that the authors acknowledge that a lack of conceptual clarity on the meaning of race and ethnicity requires clinical geneticists and biomedical researchers to ‘comprehend how, when and when not to use these socially-constructed concepts in their daily practice or research projects’ [line 599-600]. The example given is that ‘race and ethnicity’ should not be used as indicators for the presence of certain genetic variants in racialized populations but rather be used to investigate how self-identified race can lead to systemic discrimination and bias in terms of access and participation to precision oncology [line 603]. In my view, the paper would have been strengthened if this lack of conceptual clarity were highlighted earlier in the paper, and the distinctions between these terms - ‘race’ ‘ethnicity’ and ‘ancestry’ (the latter being only mentioned briefly in passing) – were made more explicit from the outset. This might then provide a more compelling basis for policy changes (e.g. calling for standardized definitions and evidence-based use of race and ethnicity (line 606), and prioritising the collection of genetic ancestry information (either during clinical consultations and in genomic databases (line 609)) rather than defaulting to use race and ethnicity as inadequate proxies.

---

## [Reviewer Report]

*Comments to Author*: This is a very precisely written summary of the literature surrounding racism in clinical research and will be a well-cited if not an essential article for those working in oncology. For these reasons, I propose the following adjustments.

1) The authors utilise the PHCRP analytical tool for this analysis and outline its principle components. It would be useful to include in the methods why this framework was selected and, in the discussion, the potential limitations or biases of it. As this review is focused around healthcrit it is important and relevant for the authors to apply a similar lens to their own practices.

2) Having summarised so many important findings in the results section, the discussion is a disappointing continuum of the introduction. Many readers will want guidance (or links to guidance) on how to change their practice but advice such as “embrace this monolithic view of racial identities” will confound. A loosening of the writing style in this section would be appropriate and should include practical guidance of how to collect racial data (for example), and linking this guidance to the 4 key points within table 4.

Overall, it makes a fascinating and humbling read.

---

## [Editor Report]

*Comments to Author*: This paper can be accepted without re-review - all items have been addressed.